# Process Reward Agents for Steering Knowledge-Intensive Reasoning

**Jiwoong Sohn** [* 1]  **Tomasz Sternal** [* 1]  **Kenneth Styppa** [* 1 2]  **Torsten Hoefler** [† 1]  **Michael Moor** [† 1]

## Abstract

Reasoning in knowledge-intensive domains remains challenging as intermediate steps are often not locally verifiable: unlike math or code, evaluating step correctness may require synthesizing clues across large external knowledge sources. As a result, subtle errors can propagate through reasoning traces, potentially never to be detected. Prior work has proposed process reward models (PRMs), including retrieval-augmented variants, but these methods operate post hoc, scoring completed trajectories, which prevents their integration into dynamic inference procedures. Here, we introduce Process Reward Agents (PRA), an inference-time method for providing domain-grounded, online, step-wise rewards to a frozen policy. In contrast to prior retrieval-augmented PRMs, PRA enables search-based decoding to rank and prune candidate trajectories at every generation step. Experiments on multiple medical reasoning benchmarks demonstrate that PRA consistently outperforms strong baselines, achieving 81.9% accuracy on MedQA with Qwen3-4B, a new state of the art at the 4B scale. Importantly, PRA generalizes to unseen frozen policy models ranging from 0.5B to 8B parameters, improving their accuracy by up to 25.7% without any policy model updates. More broadly, PRA suggests a paradigm in which frozen reasoners are decoupled from domain-specific reward modules, allowing the deployment of new backbones in complex domains without retraining. All code and data are publicly available at https://process-reward-agents.github.io/.

---

[*]Equal contribution [†]Co-last authors. [1]ETH Zürich, Switzerland [2]Heidelberg University, Germany. Correspondence to: Torsten Hoefler <torsten.hoefler@inf.ethz.ch>, Michael Moor <michael.moor@bsse.ethz.ch>.

*Proceedings of the 43rd International Conference on Machine Learning*, Seoul, South Korea. PMLR 306, 2026. Copyright 2026 by the author(s).

## 1. Introduction

Despite the success of large reasoning models (LRMs), the absence of mechanisms for validating intermediate reasoning steps remains a major challenge to reliable reasoning, particularly in high-stakes domains such as healthcare. In contrast to formal proofs or software programs, where each step can be mechanically checked against axioms, syntactic rules, or compiler constraints, medical reasoning rarely admits rigorous verification. This limitation is consequential, as clinically correct decisions must be defensible throughout the entire reasoning trace, not only in the final answer.

Establishing correctness often requires synthesizing multiple, layered sources of evidence, including primary scientific literature, clinical guidelines, and institution-specific protocols, within a landscape of knowledge that evolves continuously. Consequently, it becomes prohibitively expensive to repeatedly fine-tune each new LRM backbone to remain aligned with updated medical knowledge. Likewise, simply injecting retrieved documents into the policy context, thereby bloating it, does not guarantee that the model will attend to the right evidence at the right time, nor does it provide a mechanism to detect and correct mistakes as they emerge.

Prior work has explored the use of process reward models (PRMs) (Yun et al., 2025; Jiang et al., 2025) in medical reasoning. Med-PRM trains a process reward model for post hoc evaluation of policy-generated reasoning traces, incorporating external medical evidence via retrieval. Meanwhile, Med-S³ jointly trains a policy model and a reward model through a self-evolving framework, but does not incorporate search. Importantly, both approaches rely on post hoc evaluation, as reward signals are applied only after a complete reasoning trajectory has been generated.

This formulation limits intervention during reasoning, allowing errors to accumulate before any corrective signal is applied. Moreover, it precludes fine-grained control over the generation process, restricting the model's ability to explore alternative reasoning paths or prioritize evidence.

Building on this view, we introduce a retrieval-augmented process reward framework in which a **Process Reward Agent (PRA)** interacts with a frozen reasoning model. At each reasoning step, the PRA observes the current reason-

ing trace, optionally decides whether to search for external medical evidence, and assigns a local reward signal to guide generation in real time. This approach enables the evaluation of intermediate reasoning steps before errors propagate.

Our contributions are threefold: (i) we formulate retrieval-grounded, step-wise evaluation as an *online* control problem for medical reasoning; (ii) we propose PRA, which decouples evidence search and verification from a frozen policy to guide generation in real time; and (iii) we demonstrate that PRA enables inference-time branching and pruning strategies that generalize across tasks and backbone models.

We evaluate PRA across multiple medical reasoning benchmarks. Under a matched policy sampling budget, PRA consistently outperforms strong decoding baselines. In particular, PRA achieves 81.9% accuracy on MedQA with Qwen3-4B-Instruct, representing state-of-the-art performance at the 4B scale. Overall, these results suggest that online, step-wise rewards provide a stable and transferable mechanism for improving medical reasoning.

Additionally, we show that PRA generalizes to unseen, frozen policy models spanning 0.5B to 8B parameters, improving MedQA accuracy by up to 25.7%. These gains expose underutilized reasoning capacity in smaller models, since generation requires neither policy retraining nor context editing. Under matched policy sampling budgets, PRA continues to improve with inference-time scaling, whereas self-consistency saturates early. Ablations on reward granularity and timing indicate that these gains are driven by applying rewards at intermediate steps rather than only at completion. We detail the PRA framework and its inference-time interaction in the following sections.

## 2. Related Work

### 2.1. Medical Reasoning

Reasoning models applied in the medical domain face several domain-specific challenges. Clinically correct decisions must be grounded in both ever-expanding biomedical literature and contextual constraints such as guidelines and common practice (Norman and Eva, 2010; Fisher and Wennberg, 2003; Lu, 2011). This has motivated retrieval-based methods that provide curated and up-to-date evidence at inference time (Zakka et al., 2024; Kim et al., 2025b; Gao et al., 2026). Recent work has also introduced carefully curated retrieval corpora for targeted access, including MIRIAD (Zheng et al., 2025), as well as structured medical knowledge graphs such as MedGraphRAG (Wu et al., 2024).

In parallel, post-training has improved medical reasoning through supervised fine-tuning (SFT) and reinforcement learning from verifiable rewards (RLVR) (Chen et al., 2024;

Zhang et al., 2025a; Liu et al., 2025a; Huang et al., 2025; Thapa et al., 2025). Some systems also couple grounding and training by constructing reasoning traces from structured knowledge graphs (Wu et al., 2025).

These approaches concentrate on improving medical reasoning either through post-training or by injecting retrieved documents directly into the policy context (Lewis et al., 2020; Zakka et al., 2024; Sohn et al., 2025; Kim et al., 2025a). An alternative design, in which retrieval and evidence selection are jointly integrated with step-wise verification by a separate online controller, remains underexplored. We address this gap by decoupling retrieval from reasoning and assigning it to a process reward agent that evaluates partial reasoning traces using retrieved evidence.

### 2.2. Reward Models

Reward modeling provides an interface for allocating additional compute at inference time. Unlike outcome reward models, which score a trace solely based on its final answer, process reward models assign rewards to intermediate reasoning steps (Lightman et al., 2023). This step-level signal is particularly well suited for tree search frameworks (Liu et al., 2025b).

Training PRMs typically requires step-level supervision. Early work relied on human-annotated reasoning traces, but the high cost and limited scalability of expert annotation motivated automated alternatives. Subsequent approaches label intermediate steps using Monte Carlo rollouts from partial states, treating the fraction of correct completions as a proxy for step correctness (Wang et al., 2023). However, because models can arrive at the correct answer even when some intermediate steps are incorrect, such labels can be noisy (Zhang et al., 2025b). More recent work explores alternative supervision signals, including LLM-as-a-judge annotations (Yang et al., 2025), hybrid pipelines that combine Monte Carlo-based signals with judge-based labels (Zhang et al., 2025b), and retrieval-augmented judges that ground step evaluations in external evidence (Yun et al., 2025).

A critical challenge for process reward models is generalization across policies. Applying a PRM off-policy, that is, scoring reasoning traces generated by a policy different from the one used during training, often degrades performance due to distributional mismatch, particularly in settings where PRMs are used to guide inference-time search (Liu et al., 2025b; Snell et al., 2024). In mathematical reasoning, retrieval has been used to provide similar questions and steps as warm-up context for PRM judgment, improving generalization across models and problem types (Zhu et al., 2025).

In medicine, however, existing retrieval-augmented process reward models typically retrieve evidence only after a complete reasoning trace has been generated and apply rewards

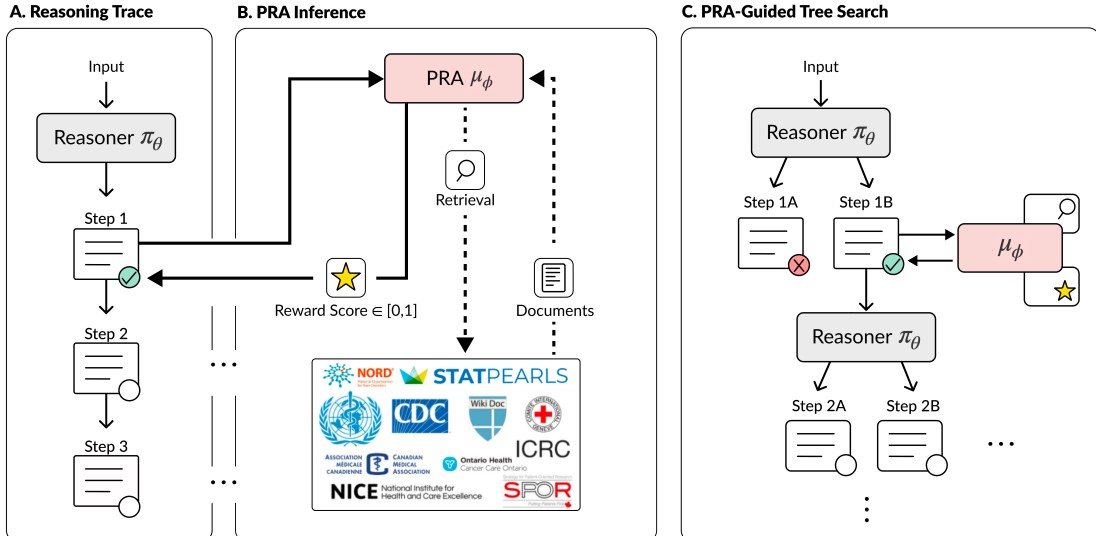

*Figure 1.* Overview of our approach. A Process Reward Agent (PRA) observes the reasoning trace generated by the frozen policy model (reasoner), decides when to search for external evidence, and assigns step-level rewards. This interaction can steer the policy at inference time, enabling more robust and controllable reasoning, particularly in knowledge-intensive domains like medicine.

post hoc (Yun et al., 2025). Consequently, online, retrieval-grounded step-wise evaluation that remains robust under policy shift remains unexplored. We address this gap by decoupling retrieval from the policy and performing search-based step-wise evaluation during generation, yielding a portable reward signal for online tree search across diverse medical reasoning policies.

## 3. Process Reward Agents

### 3.1. Problem Formulation

Let $q \in \mathcal{Q}$ denote a question, and let $y_q \in \mathcal{Y}$ denote its ground-truth answer, where $\mathcal{Q}$ and $\mathcal{Y}$ are the spaces of possible questions and answers, respectively. We assume answers are verifiable. Concretely, there exists a correctness function $C$, where $C(\hat{y}_q, y_q) = 1$ if $\hat{y}_q$ correctly matches $y_q$, and $C(\hat{y}_q, y_q) = 0$ otherwise.

Let $\pi$ be a reasoning model with frozen parameters that autoregressively generates reasoning steps. We refer to the resulting sequence as a reasoning trace $\tau = (s_1, \ldots, s_K)$. For notational simplicity, we index $\tau$ in a cumulative way, i.e., $\tau_t = (s_1, \ldots, s_t)$. Also, we define the last step $s_K$ of a completed reasoning trace to be the model's final answer.

In addition, assume access to a fixed knowledge base $\mathcal{D}$ containing domain-specific documents relevant to the questions. To make effective use of both the policy $\pi$ and the documents in $\mathcal{D}$, we aim to design a parameterized inference procedure $\mathcal{G}_\phi$ that takes as input a question, the fixed policy,

and the knowledge base, and outputs a final answer:

$$\mathcal{G}_\phi : (q, \pi, \mathcal{D}) \mapsto \hat{y}_q. \tag{1}$$

The objective is to find parameters $\phi$ that maximize the expected correctness of the produced answer:

$$\max_\phi \ \mathbb{E}_{q \sim P(\mathcal{Q}), \ \hat{y}_q \sim \mathcal{G}_\phi(q, \pi, \mathcal{D})} \Big[ C(\hat{y}_q, y_q) \Big]. \tag{2}$$

### 3.2. Process Reward Agents

We instantiate $\mathcal{G}_\phi$ as a process reward agent (PRA) that separates reasoning from evidence acquisition by delegating retrieval and evaluation to a dedicated model. The PRA consists of two components: an action controller $\mu_\phi^{\text{act}}$ and a reward scoring function $\mu_\phi^{\text{rwd}}$, both implemented as separate token-level readouts from a single model with shared parameters $\phi$. The controller observes a partial reasoning trace and selects an action:

$$\hat{a}_t \sim \mu_\phi^{\text{act}}(q, \tau_t), \ \text{where} \ \hat{a}_t \in \{\text{search, reward}\}. \tag{3}$$

When $\hat{a}_t = \text{search}$, the most relevant documents $D_t$ are retrieved from $\mathcal{D}$; when $\hat{a}_t = \text{reward}$, we set $D_t = \varnothing$. The scoring function $\mu_\phi^{\text{rwd}}$ then evaluates the most recent reasoning step $s_t$ conditioned on the partial trace $\tau_t$ and the (possibly empty) evidence set $D_t$:

$$\hat{r}_t = \mu_\phi^{\text{rwd}}(q, \tau_t, D_t). \tag{4}$$

The resulting step-wise rewards $\hat{r}_t$ steer tree search at inference time, ranking and pruning candidate trajectories online during generation.

This approach has three advantages: (1) the PRA can be trained or updated to reflect changes in the knowledge base $\mathcal{D}$ without modifying the frozen policy $\pi$, so that domain adaptation reduces to retraining a single reward module; (2) the policy is never conditioned on retrieved documents and receives no gradient signal from the PRA, different reasoning backbones can be substituted at deployment time with no retraining; and (3) conditional activation of retrieval through the controller introduces a new axis of inference-time scaling: search can be invoked selectively per step, trading off computational cost against reward signal quality within the same tree search budget.

### 3.3. PRA-Guided Tree Search

We use beam search (Boulanger-Lewandowski et al., 2012; Graves, 2012) as an inference-time-scaling method. A beam of width $B$ maintains $B$ partial reasoning traces. Each trace $\tau_t^{(j)}$ is scored by its cumulative reward:

$$R(\tau_t^{(j)}) = \sum_{i=1}^{t} \hat{r}_i^{(j)} = \sum_{i=1}^{t} \mu_\phi^{\text{rwd}}(q, \tau_i^{(j)}, D_i^{(j)}). \quad (5)$$

At every step $t$, the frozen policy $\pi$ extends each of the $B$ traces with $b$ candidate next steps (branching factor), producing $B \times b$ candidates. The PRA scores every candidate, and the top-$B$ traces by cumulative reward $R$ are retained; the rest are pruned. Generation terminates when all traces in the beam are complete, and the trace with the highest cumulative reward yields the final answer.

To enable efficient evaluation over an entire benchmark, PRA-guided tree search coordinates three models, the frozen policy $\pi$, process reward agent $\mu_\phi$, and the retriever $\rho$, across many concurrent questions, each with its own beam of traces at potentially different reasoning depths. Rather than processing questions independently, we maintain a single global queue of all active traces. At each iteration, traces are partitioned by pending stage, namely $\pi$ generation, $\rho$ retrieval, or $\mu_\phi$ evaluation (readout), and each stage is executed as a single batched operation regardless of which question, beam, or reasoning step a trace belongs to. After completion, traces re-enter the queue with updated stage tags. This synchronized stage-level batching sustains high GPU utilization even as traces become desynchronized due to variable-length reasoning, early termination, and conditional retrieval. Pseudocode is provided in Appendix Figure 6.

## 4. Experiments

### 4.1. Experimental Setup

**Datasets and Knowledge Base** We use MedQA (Jin et al., 2020) to construct the training dataset. For each question in the MedQA training split (10,178 questions), we generate eight reasoning traces using Qwen3-4B-Instruct as the

frozen reasoning model, following the policy prompt shown in Appendix Figure 7. For every partial reasoning trace $\tau_t^{(j)}$, we retrieve a corresponding set of relevant documents.

We evaluate in-distribution performance on the MedQA test split (Jin et al., 2020), ensuring that all evaluation questions are held out from the training set. To assess generalization, we further evaluate on several out-of-distribution datasets, including Medbullets (Chen et al., 2025), MedMCQA (Pal et al., 2022), MMLU-Med (Hendrycks et al., 2021; Singhal et al., 2023), GPQA (Rein et al., 2023), and clinical case datasets from The Lancet and The New England Journal of Medicine (Thapa et al., 2025).

Our knowledge base aggregates multiple medical corpora, including medical textbooks (Singhal et al., 2023), Stat-Pearls (Xiong et al., 2024), clinical practice guidelines (Chen et al., 2023), and a rare disease corpus (Wang et al., 2024).

**Retrieval** Retrieval is performed using the MedCPT dense retriever and reranker (Jin et al., 2023). For each corpus, we retrieve 200 candidate documents, rerank the combined candidate documents and retain the top 64 documents. The query used for retrieval consists of the question $q$ and the last two reasoning steps in the partial reasoning trace. This retrieval configuration is fixed and used consistently across all experiments, including training, inference, ablations, and baseline comparisons.

**Baselines** We compare against standard reasoning and retrieval baselines. Direct prompting generates answers without explicit reasoning. CoT uses Chain-of-Thought prompting to elicit step-by-step reasoning. RAG augments the input with the retrieved documents before generation. For each baseline, we also evaluate with Self-Consistency (SC), which samples multiple reasoning paths and selects the most frequent answer. For fair comparison, SC samples 64 traces, which match the compute budget of our PRA with beam search ($B = 4$, branching factor $b = 16$).

**Label Generation** We obtain reasoning and search labels for every reasoning step using Qwen3-235B-Instruct as a teacher model. Reasoning labels are generated by conditioning the teacher model on the partial reasoning trace produced by the reasoning model up to the evaluated step, together with the corresponding set of retrieved documents. For each step, we instruct the teacher model to classify the reasoning step as either correct(1) or incorrect(0) by emitting a single token (prompt in Appendix Figure 8). We use this binary output directly as the reasoning label. In addition, we extract the log-probabilities $\log p(0)$ and $\log p(1)$ assigned to the two reasoning labels, which are used for search label generation.

| Policy | Method | ID | OOD | | | | | | Average |
|---|---|---|---|---|---|---|---|---|---|
| | | MedQA | Medbullets | MedMCQA | MMLU | GPQA | Lancet | NEJM | |
| | Direct | 61.6 | 48.8 | 55.6 | 77.4 | 51.1 | 60.4 | 45.3 | 57.2 |
| | Direct + SC | 61.3 | 48.7 | 55.8 | 77.3 | 50.8 | 60.2 | 46.4 | 57.2 |
| | CoT | 72.7 | 56.5 | 61.1 | 83.7 | 60.8 | 62.4 | 62.7 | 65.7 |
| Qwen3-4B-Instruct | CoT + SC | 74.8 | 58.7 | 62.7 | 84.9 | 51.8 | 63.5 | 63.2 | 65.7 |
| | RAG | 72.2 | 55.7 | 63.3 | 85.4 | 59.2 | 62.1 | 65.7 | 66.2 |
| | RAG + SC | 76.7 | 58.4 | 64.8 | 86.2 | 54.4 | 61.0 | 66.9 | 66.9 |
| | **PRA (Ours)** | **81.9** | **65.9** | **66.2** | **86.6** | **65.1** | **70.9** | **68.0** | **72.1** |

*Table 1.* Main results on medical reasoning benchmarks. PRA outperforms direct answering, chain-of-thought (CoT), and retrieval-augmented generation (RAG) baselines on the in-distribution MedQA benchmark and six out-of-distribution benchmarks, achieving the best average score overall. Using Qwen3-4B-Instruct as the base policy model, PRA improves over the strongest baseline, RAG + SC, by 5.2 points on average.

To obtain search labels, we additionally instruct the teacher model on the same partial reasoning trace without providing any retrieved documents, while keeping the prompt structure otherwise identical. This yields a second set of log-probabilities for the same reasoning labels, enabling us to directly measure the impact of retrieval. We compute search labels using the log-probabilities obtained from these two evaluations. Intuitively, if retrieval does not affect the teacher's assessment of the reasoning step(posterior), then invoking search is unnecessary.

From a Bayesian perspective, the margin shift between two information sets measures how much the additional evidence changes the evaluator's posterior belief. Since our goal is to identify unnecessary retrieval, we treat search as necessary only when conditioning on retrieved documents induces a sufficiently large posterior update.

Let $m$ denote the margin between the log-probabilities of the correct and incorrect reasoning labels when no documents are provided, and let $m_d$ denote the corresponding margin when the teacher model is conditioned on retrieved documents:

$$m = \log p(1) - \log p(0). \qquad (6)$$

We measure the influence of retrieval on the reasoning process by computing the margin shift:

$$\Delta m = m - m_d. \qquad (7)$$

A large $|\Delta m|$ indicates that the search substantially affected the assessment of the teacher, while a small $|\Delta m|$ suggests that retrieved documents had little effect on the reasoning labels. To obtain the final binary labels, we label a step as requiring search if

$$a_t = \begin{cases} \text{search}, & |\Delta m| > \epsilon_{\text{global}}, \\ \text{reward}, & |\Delta m| \le \epsilon_{\text{global}}. \end{cases} \qquad (8)$$

where $\epsilon_{\text{global}}$ is set to the median of $|\Delta m|$ across all training data, yielding 50% of reasoning steps labeled as requiring search.

**PRA Training** We fine-tune Qwen3-4B-Instruct on the reasoning and search labels generated by the teacher model for training PRA. For each reasoning step, the model is trained to predict two binary outputs, the reasoning label and the search label. In the main experiments, we fix the search label to 1 for every step, corresponding to an always-search setting in which PRA retrieves evidence before evaluating each reasoning step, thereby ensuring maximal access to external evidence during online process-level reward guidance. For further analysis on Search–Accuracy Trade-off (Figure 3), we instead use search labels derived from the margin-shift criterion described above. This allows PRA to learn when retrieval is necessary, enabling selective search at inference time. The detailed training hyperparameters and prompt templates are provided in Appendix Section C and Section D.

**Reward Readout** Let $\ell^{(1)} \in \mathbb{R}^{|\mathcal{V}|}$ denote the logit vector at the first output slot of PRA, used for predicting the reasoning reward. The step-wise reward $\hat{r}_t = \mu_\phi^{\text{rwd}}(q, \tau_t, D_t)$ is obtained by applying a two-way softmax to the logits of tokens 0 and 1 and taking the normalized score assigned to token 1. We interpret $\hat{r}_t$ as the reward score for the correctness of the current reasoning step.

**Action Readout** Let $\ell^{(2)} \in \mathbb{R}^{|\mathcal{V}|}$ denote the logit vector at the second output slot of PRA, used for predicting the search action. The controller distribution $\mu_\phi^{\text{act}}(q, \tau_t)$ is obtained by applying a two-way softmax to the logits of tokens 0 and 1, where 1 corresponds to search and 0 corresponds to reward. We then sample the action $\hat{a}_t \sim \mu_\phi^{\text{act}}(q, \tau_t)$.

### 4.2. Results

We evaluate PRA against reasoning and retrieval baselines across seven medical benchmarks (Table 1). PRA consistently outperforms all baselines on both in-distribution and out-of-distribution benchmarks. To the best of our knowledge, our framework is the first to enable a 4B-scale model to exceed 80% accuracy on MedQA, establishing new state-

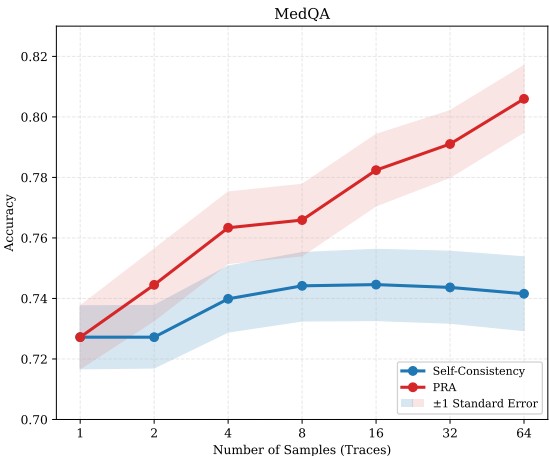

*Figure 2.* Performance on MedQA under inference time scaling. PRA continues to benefit from additional compute, while Self-Consistency saturates quickly. For SC, we estimate per-question expected accuracy via Monte Carlo sampling (1,000 trials); shaded regions show ±1 SE computed via bootstrap resampling over questions.

of-the-art performance for models of this size.

While Self-Consistency improves performance when applied to Direct, CoT, and RAG baselines on most benchmarks, we observe performance degradation with increased number of sampling on challenging benchmarks such as GPQA and Lancet. On these benchmarks, the policy model frequently produces incorrect or incomplete responses across repeated samples, causing Self-Consistency to amplify errors through majority voting. In contrast, PRA maintains stable improvements even on difficult benchmarks by guiding generation toward valid completions.

**Inference-Time Scaling Behavior** Figure 2 compares the performance of PRA and Self-Consistency as the sampling budget increases. Self-Consistency shows little improvement once the number of samples exceeds 8, whereas PRA continues to benefit from additional compute. We attribute this difference to the fact that PRA applies step-wise rewards during generation, allowing it to steer reasoning toward more promising trajectories and recover from early errors. In contrast, Self-Consistency is constrained by the policy's initial sampling distribution and can only aggregate over completed samples.

**Generalization to Unseen Datasets** PRA demonstrates strong generalization to medical reasoning benchmarks unseen during training as presented in Table 1. Across all six out-of-distribution benchmarks, PRA consistently outperforms the strongest baseline by an average of 5.2 points.

**Generalization to Unseen Policy Models** We evaluate whether PRA enables the drop-in replacement of frozen pol-

| Policy | Method | Acc. | Δ |
|---|---|---|---|
| Llama-3.1-8B-Instruct | CoT | 67.0 | – |
| | + Self-Consistency | 75.1 | +8.1 |
| | + PRA | **82.3** | **+15.3** |
| Qwen3-4B-Instruct[†] | CoT | 72.7 | – |
| | + Self-Consistency | 74.8 | +2.1 |
| | + PRA | **81.9** | **+9.2** |
| Llama-3.2-3B-Instruct | CoT | 56.0 | – |
| | + Self-Consistency | 66.2 | +10.2 |
| | + PRA | **79.1** | **+23.1** |
| Qwen2.5-3B-Instruct | CoT | 49.5 | – |
| | + Self-Consistency | 54.0 | +4.5 |
| | + PRA | **74.9** | **+25.4** |
| Llama-3.2-1B-Instruct | CoT | 36.2 | – |
| | + Self-Consistency | 44.0 | +7.8 |
| | + PRA | **61.2** | **+25.0** |
| Qwen2.5-0.5B-Instruct | CoT | 28.4 | – |
| | + Self-Consistency | 31.9 | +3.5 |
| | + PRA | **54.1** | **+25.7** |

*Table 2.* Cross-model generalization on MedQA. PRA trained with Qwen3-4B-Instruct[†] generalizes effectively to both larger and smaller policy models, with larger gains observed for smaller models. All non-daggered policies are unseen during PRA training.

icy models without requiring any retraining. Table 2 shows that PRA, despite being trained exclusively on reasoning traces from Qwen3-4B, generalizes well across a diverse set of policy models spanning both smaller and larger sizes. In several cases, models that perform poorly under standard decoding exhibit especially large relative improvements when paired with PRA, with the largest gains observed for smaller policy models. For example, on Qwen2.5-0.5B-Instruct, PRA improves MedQA accuracy from 28.4 to 54.1, corresponding to a 90.5% relative improvement over chain-of-thought.

This strong transfer is particularly notable because PRA does not modify the policy model itself. Instead, it operates purely at inference time through step-wise selection within beam search: at each step, the policy model autoregressively proposes candidate continuations, and PRA selects among them without altering the generation procedure, injecting additional context, or updating model parameters.

As a result, all generated outputs remain within the policy model's original output distribution. Unlike retrieval-augmented generation methods, which modify the model's input context, PRA exerts control solely through inference-time guidance. These results suggest that substantial gains in reasoning performance can be achieved by more effectively exploiting the latent capabilities of existing models, and that the reasoning potential of smaller policy models is considerably stronger than their standalone decoding performance may indicate.

| Reward Agent | Trained? | Search? | Method | # Sample | Acc. |
|---|---|---|---|---|---|
| $\times$ | $\times$ | $\times$ | CoT | 1 | 72.7 |
| $\times$ | $\times$ | $\times$ | CoT + SC | 64 | 74.8 |
| $\times$ | $\times$ | $\checkmark$ | RAG | 1 | 72.2 |
| $\times$ | $\times$ | $\checkmark$ | RAG + SC | 64 | 76.7 |
| Qwen3-4B | $\times$ | $\times$ | PRA | $64(B \times b)$ | 74.4 |
| Qwen3-4B | $\times$ | $\checkmark$ | PRA | $64(B \times b)$ | 76.7 |
| **PRA (ours)** | $\checkmark$ | $\checkmark$ | PRA | $64(B \times b)$ | **81.9** |

*Table 3.* Ablation study of PRA training and search on MedQA. All methods use the same frozen Qwen3-4B-Instruct policy model and differ only in whether a reward agent is used, whether it is trained, and whether search(retrieval) is enabled. Training the reward agent accounts for the majority of the performance gain, and combining it with search yields further improvements, achieving the highest accuracy with PRA.

| Method | Reward Level | Reward Time | Search | Acc. |
|---|---|---|---|---|
| CoT + SC | $\times$ | Post hoc | $\times$ | 74.8 |
| PRA (Last) | Outcome | Post hoc | $\checkmark$ | 75.7 |
| PRA (Min) | Process | Post hoc | $\checkmark$ | 74.3 |
| PRA (Max) | Process | Post hoc | $\checkmark$ | 77.5 |
| PRA (Average) | Process | Post hoc | $\checkmark$ | 77.6 |
| **PRA (Ours)** | Process | Online | $\checkmark$ | **81.9** |

*Table 4.* Ablation of outcome-level and process-level inference-time usage of the same trained PRA on MedQA. All PRA variants share identical reward model parameters; only the timing and granularity of reward application differ.

## 5. Analysis

### 5.1. Ablation on Training

Table 3 isolates the effects of reward agent, training, and search while keeping the policy model fixed to Qwen3-4B-Instruct. Under single-sample decoding, Chain-of-Thought (CoT) and retrieval-augmented generation (RAG) achieve comparable accuracy. However, increasing the number of samples yields larger gains when inference is augmented with search (i.e., retrieval augmentation), indicating that search enables more effective inference-time scaling of the policy's reasoning. We further compare the trained PRA against its untrained backbone, Qwen3-4B-Instruct, when used as the reward agent within beam search. Despite employing a different inference structure, the untrained reward agent achieves performance comparable to Self-Consistency, and when combined with retrieval, matches the accuracy of Self-Consistency with RAG. This suggests that inference-time restructuring alone is insufficient to substantially improve performance beyond the policy's native distribution, and that retrieval provides an orthogonal source of improvement.

In contrast, combining search with a trained process reward agent yields a clear additional gain. PRA, which integrates reward agent training and retrieval, achieves the highest accuracy (81.9), demonstrating that training the reward agent is critical for effective inference-time scaling with beam search and retrieval of external evidence.

### 5.2. Ablation on Inference

We further investigate whether the gains from PRA arise primarily from improved reward modeling or from how rewards are applied at inference time.

To disentangle these factors, we fix the same trained process reward agent (PRA) and vary only its inference-time usage along two axes: reward level and reward timing. Reward level specifies what is scored. Outcome-level assigns a single reward to each completed reasoning trace, whereas process-level assigns rewards to intermediate reasoning steps. Reward timing specifies when rewards are computed and used. Post hoc computes rewards only after a full trace has been generated, while online computes step-wise rewards during generation (here, within beam search), allowing them to guide reasoning as it unfolds. All settings use identical sampled traces and, when applicable, the same search mechanism; search queries are formed from the accumulated reasoning steps available at the time the reward is computed.

Table 4 shows that outcome-level PRA yields only modest improvements over Self-Consistency. For process-level, we aggregate step-wise rewards post hoc using different reduction operators (min, max, or average). This improves performance, but still underperforms settings in which rewards are applied online. Our full method, which applies step-wise rewards during generation, achieves the highest accuracy. Overall, these results indicate that the majority of the gain stems not only from stronger reward signals, but from enabling online, process-level control over the reasoning process itself.

**Search–Accuracy Trade-off** While retrieval at every reasoning step improves performance in knowledge-intensive settings, it can be costly. We therefore investigate whether PRA can learn to invoke search selectively, trading off retrieval cost against answer accuracy. To this end, we train PRA with binary search labels derived from the margin-shift criterion described in Section 4.1, enabling step-wise decisions about when retrieval is necessary. At inference time, PRA outputs a search score at each step and triggers retrieval only when this score exceeds a threshold $\theta_{\text{dep}}$. We sweep $\theta_{\text{dep}}$ from 0 to 1 in increments of 0.1 to vary the frequency of search calls. Figure 3 shows a clear trade-off between search frequency and answer accuracy: reducing search generally lowers accuracy, although the Pareto frontier indicates that selective retrieval can achieve comparable, and sometimes slightly higher, accuracy with fewer search calls.

This experiment provides further analysis of selective retrieval in PRA. In the main results (Table 1), we use an

always-search configuration, reflecting the assumption that retrieval is broadly beneficial in knowledge-intensive reasoning. This setting is well suited to knowledge-intensive evaluation and can be viewed as a practical upper bound on accuracy.

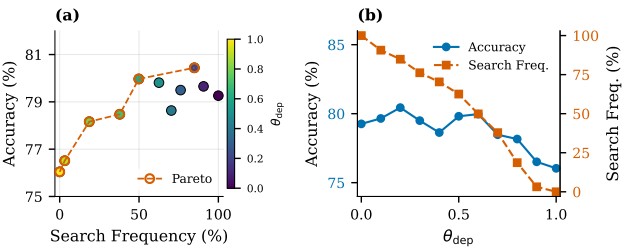

*Figure 3.* Search–accuracy trade-off on MedQA. We sweep the search threshold and report accuracy versus search frequency; the Pareto frontier highlights the best operating points for a given search budget.

### 5.3. Analysis on Margin Shift

We analyze how margin shift varies across reasoning traces on MedQA. Specifically, we compute $\Delta m$, which quantifies how the inclusion of retrieved evidence changes the teacher model's decisions between reasoning traces.

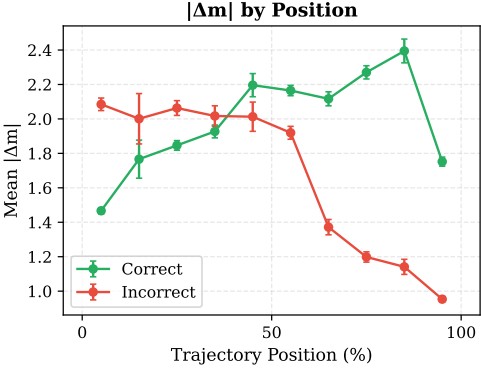

*Figure 4.* Margin shift across step positions in reasoning trajectories on MedQA, separated by traces with correct and incorrect final answers. Correct traces show larger margin shifts at later steps, whereas incorrect traces show the opposite pattern.

**Trajectory Position and Answer Correctness.** Figure 4 reports the average margin shift at different step positions within the reasoning trajectory, separated by traces with correct and incorrect final answers. We observe markedly different trends between the two groups. For traces that ultimately produce correct answers, margin shift increases toward later steps, indicating that retrieved evidence plays a larger role in the teacher model's evaluation as reasoning progresses. In contrast, for incorrect traces, margin shift decreases at later steps, suggesting that flaws in the reasoning become more apparent to the teacher model even without ad-

ditional evidence. Notably, at the final step, which typically corresponds to a concise answer selection or conclusion, retrieved evidence has little effect on margin shift, consistent with this step containing minimal substantive reasoning.

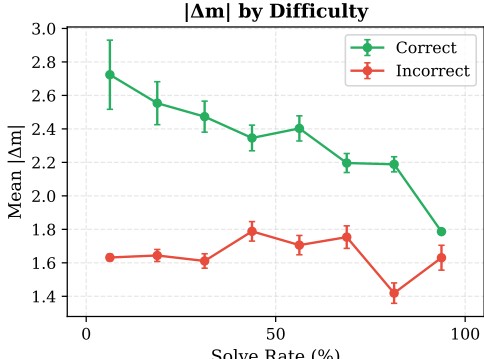

*Figure 5.* Mean absolute margin shift over reasoning steps across questions of varying difficulty, where difficulty is defined by the fraction of policy-generated reasoning samples that reach the correct answer. Correct traces consistently exhibit larger margin shifts than incorrect traces, and margin shift for correct traces is highest on harder questions and gradually decreases as solve rate increases.

**Difficulty and Answer Correctness.** Figure 5 reports margin shift across questions of varying difficulty, again separated by traces with correct and incorrect final answers. Question difficulty is defined by the fraction of reasoning samples from the policy model (Qwen3-4B-Instruct) that reach the correct answer. Consistent with the trajectory-position analysis, retrieved evidence induces larger margin shifts for correct traces, particularly on more difficult questions, while its effect remains substantially smaller for incorrect traces. One plausible interpretation could be that incorrect reasoning trajectories contain internal inconsistencies or errors that are detectable by the teacher model without strong reliance on external evidence.

## 6. Conclusion

We presented Process Reward Agents (PRA), a framework for guiding frozen reasoning models through knowledge-intensive reasoning tasks using online, step-wise, and domain-grounded process rewards. PRA reframes inference-time reasoning as a controllable search process in which a dedicated reward agent evaluates partial reasoning traces and steers generation without modifying the policy model, its parameters, or its input space. By routing retrieval and evidence usage to the reward agent rather than the policy, PRA enables fine-grained verification of intermediate steps while avoiding the sensitivity to retrieval noise and context length of standard retrieval-augmented generation. Across multiple medical reasoning benchmarks, PRA consistently outperforms strong reasoning and retrieval baselines, achieving

state-of-the-art performance for 4B-scale models on MedQA and delivering robust gains on diverse out-of-distribution datasets. We further showed that PRA generalizes across unseen policy backbones, revealing substantial underutilized reasoning capacity in smaller models. Ablation studies indicate that these gains arise primarily from applying process-level rewards online during generation rather than from post hoc scoring alone. Finally, we characterized the trade-off between retrieval cost and accuracy under selective search, showing that PRA can adaptively reduce search while preserving performance along a Pareto frontier, positioning PRA as a practical and modular approach for reliable, evidence-grounded reasoning in knowledge-intensive domains without retraining the underlying reasoning model.

## Impact Statement

This work proposes process reward agents for steering knowledge-intensive reasoning, especially medical reasoning, with frozen reasoning models. The intended impact of this approach is to increase the reliability and verifiability of reasoning traces produced by language models, with a focus on the high stakes application domain of healthcare, where individual steps need to meet a high bar to enable trust and appropriate reliance on AI systems. By explicitly rewarding the reasoning process, and grounding individual steps in the latest guidelines and literature, this work has the potential to reduce unfounded reasoning traces, catch hallucinations, and overall increase the quality of generated reasoning traces by means of domain-grounded inference-time scaling. Despite these promises, process reward agents may not fully eliminate the risk of hallucinations or remove all incorrect intermediate steps. Also, this work should be considered as a method contribution and not a ready-to-deploy system to support medical decision making. Ultimately, this work aims to make AI systems more reliable and better grounded in available external knowledge. In this sense, we hope that the presented methods and results will improve the safety of using AI in knowledge-intensive and high-stakes domains such as medicine.

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

## A. Table of Notations

Table 5 summarizes the notation used throughout the main manuscript.

| Symbol | Description |
| --- | --- |
| **System** | |
| $q$ | question |
| $\mathcal{Q}$ | space of questions |
| $\mathcal{Y}$ | space of answers |
| $y_q$ | ground-truth answer for question $q$ |
| $\hat{y}_q$ | predicted answer for question $q$ |
| $C(\hat{y}_q, y_q)$ | correctness function; 1 if $\hat{y}_q$ matches $y_q$, else 0 |
| $\pi$ | policy (frozen) |
| $\mathcal{G}_\phi$ | parameterized inference procedure $(q, \pi, \mathcal{D}) \mapsto \hat{y}_q$ |
| $\mu_\phi$ | Process Reward Agent (PRA) |
| $\mu_\phi^{\text{act}}$ | PRA controller component |
| $\mu_\phi^{\text{rwd}}$ | PRA scoring function |
| $\rho$ | retriever |
| $\mathcal{D}$ | knowledge base (collection of documents) |
| $D_t$ | set of retrieved documents at step $t$ |
| **Traces & Steps** | |
| $\tau$ | reasoning trace |
| $\tau_t$ | partial reasoning trace up to step $t$ |
| $K$ | number of reasoning steps in a trace |
| $t$ | step index $(1 \ldots K)$ |
| $s_t$ | reasoning step at position $t$ |
| $s_K$ | final step of the reasoning trace |
| $j$ | beam index |
| **PRA Actions & Rewards** | |
| $\hat{r}_t$ | predicted reward at step $t$ |
| $\hat{a}_t$ | predicted action at step $t$ |
| $r_t$ | reward label at step $t$ |
| $a_t$ | action label at step $t$ |
| $R(\tau_t^{(j)})$ | cumulative reward of partial trace $j$ up to step $t$ |
| **Training & Labels** | |
| $\ell^{(1)}$ | logit vector at the first output slot (reward) |
| $\ell^{(2)}$ | logit vector at the second output slot (action) |
| $\mathcal{V}$ | vocabulary of the PRA |
| $m$ | margin without retrieval |
| $m_d$ | margin with retrieved documents |
| $\Delta m$ | margin shift: $m - m_d$ |
| $\epsilon_{\text{global}}$ | global threshold for search labels |
| $\theta_{\text{dep}}$ | search threshold at inference |
| **Inference** | |
| $B$ | beam width |
| $b$ | branching factor |

*Table 5.* Summary of notations

## B. Stage-Level Batching

Figure 6 presents simplified pseudocode for PRA-guided beam search. Each question is managed by a `Trace` object that maintains a beam of partial reasoning traces and a stage tag $\in \{\text{REASON}, \text{REWARD}, \text{SEARCH}, \text{DONE}\}$. At each iteration, the global queue is drained, traces are partitioned by stage, and each partition is dispatched as a single batched operation to $\pi$, $\mu_\phi$, or $\rho$.

```
PRA Beam Search Pseudocode

class Trace:
    stage: REASON | REWARD | SEARCH | DONE
    beams: [{steps, cum_score, is_done}]
    candidates: [(beam_idx, text, score)]
    evidence: [str] | None

    def after_reason(self, outputs):
        self.candidates = outputs
        self.stage = REWARD

    def after_reward(self, scored):
        if should_search(scored):
            self.stage = SEARCH
        else:
            self.expand_and_prune(scored)

    def after_search(self, docs):
        self.evidence = docs
        self.stage = REWARD

    def expand_and_prune(self, scored):
        self.beams = top_B(scored, key=cum_score)
        self.stage = DONE if all_done else REASON

# ---- Global queue ----
queue = [Trace(q) for q in questions]
while queue:
    buckets = partition(queue, key=stage)
    batch_retrieve(buckets[SEARCH])
    batch_generate(buckets[REASON])
    batch_score(buckets[REWARD])
    queue = [t for t in queue if t.stage != DONE]
```

*Figure 6.* Simplified pseudocode. Each `Trace` manages one question and cycles through four stages. The global queue collects all active traces, partitions them by pending stage, and dispatches each partition as a batched operation to the policy ($\pi$), retriever ($\rho$), or reward agent ($\mu_\phi$), regardless of per-trace step index.

## C. Additional Training Details

We fine-tune Qwen3-4B-Instruct to predict the reasoning and search labels described in Section 4.1. Training is performed with a learning rate of $3 \times 10^{-5}$ using a cosine decay schedule with 100 warmup steps. We use a weight decay of 0.01, an effective batch size of 16, and train for 3 epochs in bfloat16 precision.

In the main experiments, we use the prompt shown in Figure 9 and train PRA in the always-search setting, where the search label is fixed to 1 for every reasoning step. For the Search–Accuracy Trade-off analysis, we instead train PRA using search labels derived from the margin-shift criterion described in Section 4.1.

## D. Prompt Templates

Figures 7, 8, and 9 show the prompts used throughout our experiments.

---

**Policy Prompt**

**System:**
Solve the following question step-by-step.
Do not analyze individual options in a single step.
Each step of your explanation must start with 'Step {number}:' format.
You must conclude the answer using the phrase 'the answer is (option alphabet)' at the end of your step.

**User:**
Question: {question_text}
(A): {option_A}
(B): {option_B}
. . .

Answer:

*Figure 7.* Policy prompt for frozen-reasoner generation during PRA-guided beam search. The system message requires explicit step-wise reasoning ('Step {number}:') and a standardized final-answer phrase for parsing; the user message lists the question as 'Question: . . .' with '(letter): text' options and ends with 'Answer:'. Retrieved documents are not shown to the policy.

---

**Teacher Prompt**

**System:**
You are a medical expert responsible for evaluating the quality of the last reasoning step in a solution to a medical question.
You are provided with relevant documents, the question, and the reasoning trace including prior steps.
Your task is to critically assess only the last reasoning step, considering its logical coherence, medical validity, and consistency with the evidence.

You must only return one score, and output nothing else:
Reasoning Score: Score 1 if the last step is logically coherent, medically sound, and aligns with the provided evidence; otherwise, score 0.

Output only your reasoning score in the following format:

[REASONING_SCORE]: 1 or 0 (1: correct, 0: incorrect)

**User:**
=== DOCUMENTS ===
Doc 1: {document_1}
. . .

=== QUESTION ===
{question_text}

(A): {option_A}
(B): {option_B}
. . .

=== CORRECT ANSWER ===
({answer_idx}): {answer_text}

=== REASONING TRACE ===
Step 0: {step_0_text}
Step 1: {step_1_text}
. . .

*Figure 8.* Teacher prompt for step-level reasoning labels. The user message provides retrieved documents, the question with '(letter): text' options, the correct answer, and the reasoning trace through the current step; the model assesses only the last step and returns a single score as [REASONING_SCORE]: 0 or 1. The documents block is omitted in a second pass to estimate the no-retrieval margin used for search-label construction.

---

**PRA Prompt**

**System:**
You are an evaluator responsible for assessing the quality of the last reasoning step in a solution to a medical question.
You are provided with relevant documents, the question, and the reasoning path (including prior steps and their rewards, if they exist).
Your task is to critically assess only the last reasoning step, considering its logical coherence, medical validity, and consistency with the evidence.

You must only return two scores, and output nothing else:
1. Reasoning Reward: Score 1 if the last step is logically coherent, medically sound, and aligns with the provided evidence; otherwise, score 0.
2. Search Reward: Score 1 if, in order to evaluate the last reasoning step, you needed to refer to the provided evidence (i.e., the step required searching for or validating with external information), or if the reasoning step itself explicitly involves searching, retrieval, or referencing outside knowledge; otherwise, score 0.

Provide your evaluation as two numbers, separated by a comma and a space, with no additional explanation or text. The first number is the Reasoning Reward, and the second is the Search Reward, as in the following examples:
0,0
1,0
0,1
1,1

For instance:
- If Reasoning Reward = 0 and Search Reward = 1, write: 0,1
- If both are 1: 1,1
- If both are 0: 0,0
- If Reasoning Reward = 1 and Search Reward = 0: 1,0

**User:**
=== DOCUMENTS ===
Doc 1: {document_1}
. . .

=== QUESTION ===
{question_text}

(A): {option_A}
(B): {option_B}
. . .

=== REASONING SOLUTION ===
{prior_steps}
{current_candidate_step}

*Figure 9.* PRA prompt for beam-search inference and supervised fine-tuning. The documents block is included only on the post-retrieval reward pass.

