# OpenReview forum: "Process Reward Agents for Steering Knowledge-Intensive Reasoning"
_ICML.cc/2026/Conference — ICML 2026 regular_

### Official Review · Reviewer_DRkF · 2026-03-11

**Soundness:** 4
**Presentation:** 3
**Significance:** 3
**Originality:** 3
**Overall Recommendation:** 5
**Confidence:** 5

**Summary:**

This paper proposes Process Reward Agents (PRA), an online, retrieval-grounded process reward framework that steers frozen reasoning models during inference. A trained reward agent assigns step-wise scores and optionally triggers retrieval, guiding beam search to branch and prune trajectories at each step. On multiple medical QA benchmarks, PRA improves over strong CoT, RAG, and self-consistency baselines, and shows notable cross-policy gains without updating the policy. Ablations attribute most of the gains to applying process-level rewards online, rather than post hoc.

**Compliance With Llm Reviewing Policy:**

Affirmed.

**Final Justification:**

My assessment of the paper remains positive. I am keeping my assessment given the reply of the authors.

**Key Questions For Authors:**

1. Why does the teacher prompt include the correct answer when labeling step correctness? Can you provide some explanation about the impact on PRA performance and on label quality (e.g., agreement with human raters)?
2. Do you have any estimation on the average wall-clock latency per question, number of retrieval calls, tokens processed?

**Limitations:**

yes

**Strengths And Weaknesses:**

### Strengths

1.  The idea of introducing an online, retrieval-grounded process reward controller that decouples evidence acquisition from a frozen policy and applies step-wise rewards during generation is inspiring.
2.  The thorough ablations contrasting outcome-level vs. process-level and post hoc vs. online reward usage isolate where improvements arise makes the design choice more convincing.
3.  The role of retrieval and the trade-off between search budget and accuracy are explicitly explored.

### Weaknesses

1. The bootstrap data for the PRA is a distilled dataset of Qwen-3 235B. No human or expert validation of PRM step labels or of the PRA’s step-wise judgments; correlation to expert annotations would strengthen the claim of medically meaningful step verification.

2. Compute fairness is not fully addressed: PRA may invoke retrieval at many intermediate steps across multiple branches, likely incurring much higher latency and retrieval cost than SC or standard RAG. Wall-clock time, retrieval-call counts, and token usage are not reported.
3. On the presenting aspect: the definition and delimitation of “reasoning steps” for branching are under-specified (e.g., how the model samples b “candidate next steps,” step extraction boundaries, and the handling of partial step tokens).

---

> ### Author Rebuttal · Authors · 2026-03-31
>
> Dear Reviewer DRkF,
>
> We thank the reviewer for the thoughtful feedback and address each point concisely below.
>
> > W1: PRA is distilled from Qwen3-235B, without expert validation.
>
> We agree that expert validation would strengthen the claim that the learned step-level judgments are medically meaningful. In the current paper, Qwen3-235B is used only to generate step-level supervision: binary correctness labels for intermediate steps and the margin-shift signal for retrieval dependence. Our claim is therefore not that the teacher provides gold annotations, but that it provides a scalable supervisory signal for training a smaller Process Reward Agent.
>
> We chose an open-source teacher for reproducibility, since closed API-based models are difficult to use as stable references because their behavior may change over time without versioned documentation. To assess sensitivity to teacher choice, we additionally performed cross-teacher agreement analysis across Qwen3-235B, Gemini-2.5-Flash, Gemini-3-Flash-Preview, GPT-5.4-mini, and o4-mini. Agreement is consistently substantial, typically in the 80 to 90% range:
>
> |Teacher|qwen|g2.5|g3|gpt5.4m|o4m|
> |-|-:|-:|-:|-:|-:|
> |**qwen**|—|79.4|78.1|80.1|80.7|
> |**g2.5**|79.4|—|89.7|80.5|88.2|
> |**g3**|78.1|89.7|—|80.2|90.5|
> |**gpt5.4m**|80.1|80.5|80.2|—|82.4|
> |**o4m**|80.7|88.2|90.5|82.4|—|
>
> This suggests that the supervision signal is not idiosyncratic to one teacher, though it does not replace direct expert validation. We will revise the manuscript to clarify this scope.
>
> > W2 / Q2: Compute fairness is not fully addressed. Do you have estimates of latency, retrieval calls, and tokens?
>
> We agree that compute should be reported explicitly. We therefore added both token-based and wall-clock analyses.
>
> |Method|Toks/q|x CoT@1|Acc.|Gain|
> |-|-:|-:|-:|-:|
> |CoT@1|665|1.0x|72.28|—|
> |CoT-SC@64|42582|64.0x|75.47|+3.19|
> |Beam(log-prob)|29521|44.4x|71.07|-1.21|
> |Beam(length-norm)|49833|74.9x|75.16|+2.88|
> |**Beam+PRA**|**33046**|**49.7x**|**81.45**|**+9.17**|
>
> |Method|Gain|Extra toks|Gain / 10k toks|
> |-|-:|-:|-:|
> |CoT-SC@64|+3.19|41917|0.76|
> |Beam(length-norm)|+2.88|49168|0.59|
> |**Beam+PRA**|**+9.17**|**32381**|**2.83**|
>
> Thus, PRA is not the cheapest decoding strategy, but it is substantially more token-efficient than scaling self-consistency or using reward-free beam search.
>
> We also report wall-clock latency and retrieval calls using our batched implementation:
>
> |Setting|Acc.|Time/q|Retrieval/q|
> |-|-:|-:|-:|
> |theta_dep=0.0|79.26|46.4s|5.42|
> |theta_dep=0.2|80.44|66.8s|4.62|
> |theta_dep=1.0|76.04|32.0s|0.00|
>
> The best selective-search setting achieves the highest accuracy but is slower because it performs both search deliberation and retrieval for most steps. We will revise the paper to present PRA as a method with a clear and tunable performance-compute trade-off.
>
> > W3: Reasoning-step construction for branching is under-specified.
>
> We agree and will clarify the procedure. PRA branches at the step level, not the token or sentence level. The reasoner is prompted to emit explicit step headers (e.g., `Step 1:`), and we sample `b` candidate next steps by decoding until the next step boundary. Boundaries are detected with step-specific stop sequences and parsed with a regular expression; if a partial next-step header is emitted, it is removed from the current step and prepended to the next call. We will revise the manuscript to describe this explicitly.
>
> > Q1: Why does the teacher prompt include the correct answer when labeling step correctness?
>
> We agree that including the correct answer introduces hindsight information, and we will clarify this design choice. Our intention is to use the teacher as a verifier of step quality, not as a simulator of the test-time policy’s information state. Providing the gold answer lets the teacher judge whether an intermediate step supports a trajectory leading to the correct solution. We therefore view these labels as ground-truth-referenced verification signals rather than as a perfect model of what the policy itself could infer at that point.
>
> This choice is motivated by the fact that step correctness is often ambiguous without reference to the final target: a step may seem locally plausible while still steering the reasoning toward an incorrect conclusion. Conditioning on the gold answer gives a clearer verification signal for the Process Reward Agent to learn. At the same time, we agree that this design may affect both label characteristics and downstream performance. To address this, we will add a comparison using a teacher prompt without access to the gold answer. Agreement with human raters would also be valuable future work. Our current claim is therefore narrower: teacher-generated labels are a practical supervisory signal, not a substitute for formal expert annotation.
>
> We thank the reviewer again for the thoughtful and helpful feedback, and we will incorporate these improvements in the revision.

---

> > ### Author Rebuttal · Reviewer_DRkF · 2026-04-08
> >
> > My concerns are adequately addressed.

---

### Official Review · Reviewer_GwDA · 2026-03-11

**Soundness:** 3
**Presentation:** 3
**Significance:** 3
**Originality:** 3
**Overall Recommendation:** 4
**Confidence:** 3

**Summary:**

This paper introduces Process Reward Agents (PRA): a test-time reward agent decoupled from the policy model that can retrieve external medical knowledge to perform online, step-by-step evaluation of the reasoning process, with rewards directly used for branching and pruning in beam search. Unlike existing medical PRMs that mostly perform post hoc scoring after complete trajectory generation, the core advancement of this work lies in moving "retrieval + process reward" forward into the generation process, thereby enabling real-time control over intermediate steps. The paper achieves 80.9% on MedQA with Qwen3-4B-Instruct and reports significant generalization gains across multiple medical OOD benchmarks and unseen policy models (0.5B–8B). Through ablations, the authors demonstrate that performance improvements primarily stem from the online, process-level usage of rewards rather than merely stronger post-hoc scorers.

**Compliance With Llm Reviewing Policy:**

Affirmed.

**Key Questions For Authors:**

1. You use margin shift from the teacher under "with/without retrieval" conditions and threshold by global median to define whether search is needed. What evidence demonstrates consistency between this label and "genuine need for external knowledge validation"? Would conclusions remain stable with small-scale human annotation or alternative thresholding strategies?

2. Have you compared against stronger baselines closer to PRA's inference morphology, such as: (i) using only the trained reward model for post-step reranking of beam candidates without making search decisions; (ii) beam search with fixed per-step retrieval; (iii) step-level reranking based on heuristics or LLM judges?

3. PRA performs well on unseen policies. How substantial are the differences in reasoning trace styles across these models? Have you analyzed PRA's calibration on different models (e.g., correspondence between reward scores and final accuracy)?

4. Are results sensitive to knowledge base sources, retriever/reranker choices, and top-k document counts? Are there cases where erroneous retrieval significantly misleads PRA?

5. The paper primarily validates on multiple-choice medical QA. Do the authors have evidence that this method transfers to more open-ended clinical reasoning tasks without option constraints?

**Strengths And Weaknesses:**

## Strengths

1. The paper identifies a genuine pain point in medical reasoning—intermediate steps are often not locally verifiable—and reframes retrieval-augmented process rewards from traditional post-hoc evaluators to online controllers, enabling real-time intervention of external knowledge validation during generation. This paradigm shift (post hoc → online) is not merely a technical adjustment but redefines the role of process reward models in reasoning, carrying clear methodological value.

2. Table 4 effectively isolates two factors—"stronger reward model itself" versus "better usage of rewards"—by fixing the same trained PRA and varying only the granularity (outcome vs process) and timing (post hoc vs online) of reward application. This experiment directly supports the core claim that performance gains stem primarily from online process-level control rather than simply stronger post-hoc scorers, representing the most critical evidence chain in the paper.

3. Trained solely on Qwen3-4B trajectories, PRA zero-shot transfers to multiple unseen models including Llama-3.1-8B and Qwen2.5-0.5B, with more pronounced gains for smaller models (up to +25.7%). If robust, this result substantiates the methodological significance of "decoupling reward modules from reasoning backbones" and offers a potentially more efficient alternative to per-model post-training for rapid adaptation to new backbones.

4. Figure 3 demonstrates a Pareto frontier achieved by adjusting the search threshold θ_dep, showing that selective retrieval can maintain high performance while reducing computational costs, providing actionable room for cost-benefit trade-offs in practical deployment.

## Weaknesses

1. Search labels are generated via margin shift (∆m) under "with/without retrieval" conditions from the teacher model, thresholded by global median—essentially a proxy for "teacher sensitivity to documents" rather than reliable supervision for "step genuinely requires external knowledge." This definition carries three risks: (1) teacher judgments may be confounded by document noise or prompt formatting; (2) the forced 50% ratio is human-calibrated rather than task-derived; (3) consistency with human expert judgment remains unvalidated.

2. PRA (beam search + process reward + selective retrieval) and Self-Consistency are not symmetric in sample utilization. Key baselines are missing: (1) beam search with fixed per-step retrieval; (2) post-step reranking using only the trained reward model without learning search decisions; (3) reranking baselines based on simple heuristics. Current results demonstrate PRA's superiority over standard decoding, but are insufficient to fully establish that "gains come from PRA-specific design rather than merely more complex search control structures."

3. The paper claims PRA's "outputs remain strictly within the original policy distribution," yet beam pruning substantially changes the induced output distribution by altering sampling paths. Moreover, cross-model generalization remains limited to similar instruction-tuned open-source LLM families and the same multiple-choice QA format, with no validation of transferability to different reasoning styles or heterogeneous task modalities.

4. The method heavily relies on fixed knowledge bases and MedCPT retrieval configurations, but lacks sensitivity analysis for: (1) knowledge base source composition; (2) retriever/reranker choices; (3) top-k document count variations; (4) misleading risks from erroneous or outdated retrieval.

5. Experiments focus on multiple-choice medical QA, yet real clinical decision-making often involves open-ended reasoning and dynamic interaction. The extrapolation of conclusions toward "reliable medical reasoning" is somewhat overstated; the current validation scope should be more explicitly delimited.

---

> ### Author Rebuttal · Authors · 2026-03-31
>
> Dear Reviewer GwDA,
>
> We thank the reviewer for the thoughtful feedback and address each point briefly below.
>
> > Weakness 1 / Question 1: Do margin-shift search labels reflect genuine need for external knowledge?
>
> We agree that this label should not be framed as a gold annotation of intrinsic search necessity. Our intent is narrower: the label is a proxy for retrieval dependence, namely whether retrieved evidence causes a meaningful counterfactual change in the teacher’s evaluation of a step. From a Bayesian perspective, the margin shift reflects how much the additional evidence updates the evaluator’s posterior belief. We will revise the paper to make this distinction explicit. We also note that our conclusions do not depend on the global median threshold alone: Figure 3 already sweeps thresholds across a wide range of search rates. Directly asking the teacher whether search is needed produced an unusably imbalanced label distribution of 99.5% of steps marked positive, which motivated the margin-shift formulation. We agree that small-scale human validation would be valuable future work.
>
> > Weakness 2 / Question 2: Do gains come from PRA itself, rather than a more complex search controller?
>
> We agree that stronger nearby baselines are important. The current paper already includes ablations(table 4) that isolate reward timing and granularity, and we additionally compare against heuristic beam-search baselines. The results consistently show that neither beam search alone nor post-hoc reward reranking recovers the full gain of online process-level control.
>
> Regarding fixed per-step retrieval, Figure 4 already sweeps the search threshold up to near-always-search regimes, showing that the benefit is not tied to a single search budget.
>
> > Weakness 3: Framing around “original policy distribution” and breadth of generalization.
>
> We agree that “strictly within the original policy distribution” is too strong. PRA does change the induced output distribution through pruning and path selection. The precise claim is that PRA does not modify policy parameters and does not inject external feedback tokens into the reasoner’s context; it only selects among continuations proposed by the frozen policy itself. We will revise the wording accordingly in the revision.
>
> We also agree that our generalization claim should be narrower. The main evidence is transfer across unseen frozen policies in knowledge-intensive question answering. To partially broaden this, we additionally evaluate on HotpotQA fullwiki in weakness5/question5, which changes domain, corpus, and answer format.
>
> > Weakness 4 / Question 4: Sensitivity to retrieval setup.
>
> Our claim is not that one MedQA retrieval pipeline is universally optimal, but that PRA is modular and can benefit from curated external corpora without relying on one retrieval configuration. We therefore added corpus and top-K ablations:
>
> |Removed corpus|Acc.|Delta|
> |-|-:|-:|
> |None(full)|81.76|0.00|
> |CPG|82.39|+0.63|
> |Rare disease|81.76|0.00|
> |Textbooks|81.13|-0.63|
> |StatPearls|81.13|-0.63|
>
> |Top-K|Acc.|Delta|
> |-|-:|-:|
> |16|78.93|-2.83|
> |32|80.19|-1.57|
> |64|81.76|0.00|
>
> These results suggest that performance changes only modestly under leave-one-out ablations and degrades gradually as K decreases. We agree that sensitivity to retriever/reranker choice and stress tests with intentionally misleading retrieval are important future directions.
>
> > Weakness 5 / Question 5: Does PRA transfer beyond multiple-choice, medical QA?
>
> We agree that the present paper does not establish transfer to full clinical decision-making or unconstrained open-ended medical generation, and we will delimit that scope more clearly. To test whether PRA is tied to answer-option constraints, we additionally evaluate on HotpotQA fullwiki:
>
> |Method|EM|F1|Delta EM|Delta F1|
> |-|-:|-:|-:|-:|
> |CoT@1|19.6|28.0|-|-|
> |CoT-SC@64|20.6|28.8|+1.0|+0.8|
> |**PRA**|**25.2**|**35.5**|**+5.6**|**+7.5**|
> |RAG@1|48.8|63.2|-|-|
> |RAG-SC@64|50.4|65.1|+1.6|+1.9|
> |**RAG-PRA**|**51.6**|**66.9**|**+2.8**|**+3.7**|
>
> This provides supporting evidence that PRA is not inherently tied to multiple-choice decoding, although we do not claim that it establishes open-ended, interactive clinical setting transfer.
>
> > Question 3: Is PRA calibrated across unseen policies with different trace styles?
>
> To probe this, we tested whether the normalized cumulative reward score of the traces predicts final correctness across different reasoners using logistic regression. The relationship is consistently positive and significant:
>
> |Model|F1|AUC|Beta|p-value|
> |-|-:|-:|-:|-:|
> |Qwen3-4B|0.868|0.775|5.939|<0.05|
> |Qwen3-8B|0.906|0.846|8.108|<0.05|
> |Qwen3-30B|0.893|0.849|7.936|<0.05|
>
> Thus, higher PRA scores remain meaningfully associated with better final outcomes across unseen policies, despite differences in scale and likely trace style. We will include this calibration analysis in the appendix and revise the framing throughout to better match the actual scope of the evidence.

---

> > ### Author Rebuttal · Reviewer_GwDA · 2026-04-02
> >
> > Thank you for your reply, which has solved my problem. I will keep my score.

---

### Official Review · Reviewer_R9nZ · 2026-03-12

**Soundness:** 2
**Presentation:** 2
**Significance:** 2
**Originality:** 3
**Overall Recommendation:** 3
**Confidence:** 4

**Summary:**

This paper proposes Process Reward Agents (PRA), a test-time framework designed to steer frozen reasoning models in knowledge-intensive tasks. The authors focus on medical reasoning, a domain where intermediate steps often lack verifiability and require the external knowledge sources. PRA decouples the reasoning from evidence retrieval and verification by dynamically deciding when to retrieve documents and assigning rewards to steps. Experimental results show that the method achieves SOTA performance on MedQA for a 4B model (80.9%) and demonstrates generalization across OOD medical benchmarks.

**Compliance With Llm Reviewing Policy:**

Affirmed.

**Key Questions For Authors:**

- Do authors have any analysis beyond the Qwen3-4B-Instruct model as PRA? If not, then it is very hard to prove generalizability of claims.

**Limitations:**

Yes

**Strengths And Weaknesses:**

Strengths:

- Decoupling retrieval and verification from the policy is novel and interesting.
- Focusing on medical problems, which is a high-stakes domain, is a very valuable study.
- Empirical results on MedQA is impressive (+25.7% gain)
- I really like the plug-and-play nature of PRA, i.e., PRA trained on Qwen3-4B improves performance across different models.

Weaknesses:

- A major concern is the heavy reliance on labels generated by the Qwen3-235B teacher model. It is unclear how sensitive the framework's success is to the specific model, and whether utilizing stronger models like GPT or Gemini would help more or less?
- The generalizability of the PRA approach beyond the medical domain remains another major issue. Evaluating the method on standard reasoning benchmarks such as MATH, or MMLU would provide better indication of the approach's utility.
- Another concern is the scalability of this method. Since PRA relies heavily on labeled data for training, it is questionable whether the approach can effectively scale to general-domain tasks where the training data is massive and diverse. Also, generating labels might not be easy.
- Authors use a specific retrieval source for MedQA, the feasibility of such retrieval when generalizing across the general domain or even other tasks in the medical domain is not discussed in the paper.
- Another major concern is the latency (maybe time or tokens vs. performance). The authors have not reported any latency measures compared to CoT or RAG; whether the performance gains are significant enough to justify the additional computational or not.

---

> ### Author Rebuttal · Authors · 2026-03-31
>
> Dear Reviewer R9nZ,
>
> We thank the reviewer for the insightful review and address each point below.
>
> > Question 1: Do authors have analysis beyond using Qwen3-4B-Instruct as the PRA?
>
> To address this directly, we additionally trained a smaller PRA with Qwen3-1.7B while keeping the policy fixed to Qwen3-4B-Instruct. It still reaches 77.67%, outperforming both CoT-SC@64 and reward-free beam search. This suggests that PRA is not tied to one specific reward-model backbone.
>
> |Method|PRA Backbone|Acc.|Δ|
> |-|-|-:|-:|
> |CoT-SC@1|-|72.28%|-|
> |CoT-SC@64|-|75.47%|+3.19|
> |Beam(length-normalized log-prob)|-|75.16%|+2.88|
> |Beam+PRA|Qwen3- 1.7B|77.67%|+5.39|
> |Beam+PRA|Qwen3- 4B -Instruct|81.76%|+9.48|
>
> Table 1 also shows results across unseen benchmarks, and Table 2 across unseen policy models. The additional Qwen3-1.7B result strengthens robustness across PRA backbones as well.
>
> > Weakness 1: How sensitive is PRA to the Qwen3-235B teacher? Would GPT or Gemini help more or less?
>
> Qwen3-235B-Instruct is used only to label intermediate steps and define the retrieval-necessity signal during PRA training. We chose an OSS model for reproducibility.
>
> While we could not retrain PRA with multiple teachers before the rebuttal deadline, we performed label-agreement analysis across Qwen3-235B, Gemini-2.5-Flash, Gemini-3-Flash-Preview, GPT-5.4-mini, and o4-mini. Pairwise agreement is typically in the 80–90% range:
>
> |Teacher|qwen|g2.5|g3|gpt5.4m|o4m|
> |-|-:|-:|-:|-:|-:|
> |**qwen**|—|79.4|78.1|80.1|80.7|
> |**g2.5**|79.4|—|89.7|80.5|88.2|
> |**g3**|78.1|89.7|—|80.2|90.5|
> |**gpt5.4m**|80.1|80.5|80.2|—|82.4|
> |**o4m**|80.7|88.2|90.5|82.4|—|
>
> Consensus is also high: 64.4% of steps have unanimous agreement across all five teachers, 22.7% have only one outlier, and only 12.4% show a true 3-vs-2 split. This suggests the supervision signal is not an idiosyncratic artifact of Qwen3-235B. A full teacher-swap retraining study would be valuable and we will include it in the camera-ready version if accepted.
>
> > Weakness 2/3: What about generalizability beyond medicine and scalability to general-domain tasks?
>
> Our goal is not to improve frozen reasoners on general benchmarks, but to adapt them to knowledge-intensive settings with verified external corpora. To test generalizability beyond medicine, we additionally evaluate on HotpotQA FullWiKi.
>
> |Method|EM|F1|Δ EM|Δ F1|
> |-|-:|-:|-:|-:|
> |CoT@1|19.6|28.0|-|-|
> |CoT-SC@64|20.6|28.8|+1.0|+0.8|
> |**PRA**|**25.2**|**35.5**|**+5.6**|**+7.5**|
>
> We also test a setting where the reasoner itself is given retrieved documents:
>
> |Method|EM|F1|Δ EM|Δ F1|
> |-|-:|-:|-:|-:|
> |RAG@1|48.8|63.2|-|-|
> |RAG-SC@64|50.4|65.1|+1.6|+1.9|
> |RAG+PRA|**51.6**|**66.9**|**+2.8**|**+3.7**|
>
> These results suggest that PRA’s benefit is not purely domain-specific and remains complementary to retrieval. We agree that scaling to unconstrained general-domain tasks would be challenging, but that is out of our research scope; our focus is high-stakes, knowledge-intensive domains where verified corpora and step-level supervision are feasible.
>
> > Weakness 4: Is the retrieval setup too specific to the MedQA source configuration?
>
> Our claim is not that one fixed MedQA retrieval setup transfers unchanged to all tasks. Rather, PRA is modular: the policy remains frozen, and adaptation is achieved by changing the external corpus used by the reward agent. The HotpotQA experiment follows this recipe.
>
> To address robustness, we added retrieval ablations over corpus composition and retrieval depth:
>
> |Removed corpus|Acc.|Δ|
> |-|-:|-:|
> |None (full)|81.76|0.00|
> |clinical practice guidelines|82.39|+0.63|
> |rare disease corpus|81.76|0.00|
> |textbooks|81.13|-0.63|
> |StatPearls|81.13|-0.63|
>
> |Rerank top-K|Acc.|Δ|
> |-|-:|-:|
> |16|78.93|-2.83|
> |32|80.19|-1.57|
> |64 (default)|81.76|0.00|
>
> Performance changes only modestly under leave-one-out ablations and degrades gradually as K is reduced, suggesting that PRA is not brittle to one indispensable source or one narrowly tuned retrieval setting.
>
> > Weakness 5: Are the gains large enough to justify the latency / compute cost?
>
> We agree that compute should be reported explicitly. We therefore added both token-based and wall-clock analyses.
>
> |Method|Est. toks/q|x CoT@1|Acc.|Gain|
> |-|-:|-:|-:|-:|
> |CoT@1|665|1.0x|72.28|—|
> |CoT-SC@64|42582|64.0x|75.47|+3.19|
> |Beam(log-prob)|29521|44.4x|71.07|-1.21|
> |Beam(length-norm)|49833|74.9x|75.16|+2.88|
> |**Beam+PRA**|**33046**|**49.7x**|**81.45**|**+9.17**|
>
> |Method|Gain|Extra toks|Gain / 10k toks|
> |-|-:|-:|-:|
> |CoT-SC@64|+3.19|41917|0.76|
> |Beam(length-norm)|+2.88|49168|0.59|
> |**Beam+PRA**|**+9.17**|**32381**|**2.83**|
>
> Thus, PRA is not the cheapest decoding strategy, but it is substantially more compute-efficient than scaling self-consistency or using reward-free beam search.
>
> |Setting|Acc.|Time/q|
> |-|-:|-:|
> |theta_dep=0.0|79.26|46.4s|
> |theta_dep=0.2|80.44|66.8s|
> |theta_dep=1.0|76.04|32.0s|
>
> This shows a clear and tunable performance-compute trade-off, which we will report explicitly in the revision.

---

> > ### Author Rebuttal · Reviewer_R9nZ · 2026-04-04
> >
> > Thanks for the rebuttal. Please add additional experiments to other comments in the revised version. But some of my concerns still remain, including very key concern regarding reliance on Qwen3-235B.
> >
> > 1. While the label-agreement is helpful, without actually training PRA with a different teacher, the claim that the framework isn't overly reliant on the artifacts of Qwen3-235B is hard to justify. I agree that it is not feasible to provide experiments during the scope of rebuttal.
> > 2. For Q1, authors used same family model which answered my concern partially, but it is still remain a question about generalizability beyond Qwen models.
> >
> > Thus, I am keeping my score.

---

> > > ### Author Response · Authors · 2026-04-08
> > >
> > > Dear Reviewer R9nZ,
> > >
> > > We thank the reviewer again for the thoughtful follow-up. We agree that our previous rebuttal may have partially addressed the concern about the generalizability of PRA reward backbones.
> > >
> > > To address this directly, we additionally trained PRA with **non-Qwen reward backbones**, namely **Llama-3.1-8B-Instruct** and **MedGemma-4B**, while keeping the rest of the experimental setup unchanged.
> > >
> > > We compare these against the same baselines:
> > >
> > > |Method|PRA backbone|Accuracy|Δ vs CoT|Δ vs CoT-SC@64|Δ vs Beam (log-prob)|Δ vs Beam (length-norm)|
> > > |-|-|-:|-:|-:|-:|-:|
> > > |CoT|-|72.28|-|-|-|-|
> > > |CoT-SC@64|-|75.47|+3.19|-|+4.40|+0.31|
> > > |Beam (log-prob)|-|71.07|-1.21|-4.40|-|-4.09|
> > > |Beam (length-normalized log-prob)|-|75.16|+2.88|-0.31|+4.09|-|
> > > |Beam + PRA|**Qwen3-1.7B**|**77.67**|**+5.39**|**+2.20**|**+6.60**|**+2.51**|
> > > |Beam + PRA|**Llama-3.1-8B-Instruct**|**78.00**|**+5.72**|**+2.53**|**+6.93**|**+2.84**|
> > > |Beam + PRA|**MedGemma-4B**|**78.62**|**+6.34**|**+3.15**|**+7.55**|**+3.46**|
> > > |Beam + PRA|Qwen3-4B-Instruct|81.76|+9.48|+6.29|+10.69|+6.60|
> > >
> > > These results show that PRA is **not limited to Qwen-based reward models**. Even when the reward agent is trained with backbones from different model families and of different sizes, it consistently improves the same frozen Qwen3-4B-Instruct policy over strong baselines. In these experiments, both the frozen policy and the training traces are based on **Qwen3-4B-Instruct**. The matched **Qwen3-4B-Instruct** PRA backbone achieves the strongest result as reported in our manuscript, but notably, reward backbones from other model families still deliver consistent gains over the same strong baselines. We will add these results in the revised version.
> > >
> > > More broadly, these experiments complement the paper’s evidence for **policy generalization**(PRA generalizes to 1 + 7 unseen frozen policy models) and **benchmark generalization**(1 + 6 unseen benchmark datasets) by adding a third axis, **backbone generalization**(1 + 3 backbone models). In our submitted codebase, the same end-to-end pipeline can be reused for training, inference, and evaluation simply by changing the model name. We therefore believe these results are not merely isolated examples, but evidence that the framework is practically compatible with a range of backbone choices. If the reviewer or future readers would find additional backbone comparisons helpful, they can be incorporated readily within the same setup, and we would be very happy to experiment and include more models to further clarify and strengthen the paper.
> > >
> > > Regarding the reviewer’s concern about reliance on Qwen3-235B as the teacher, we agree that label agreement alone may not fully establish insensitivity to teacher choice, and we will revise the text to make this limitation explicit. At the same time, we would like to clarify our perspective on teacher selection. We deliberately chose an **open-weight teacher**, which we view as an important practical strength of the paper. It shows that the entire PRA pipeline can be implemented without relying on proprietary API models such as GPT, while still achieving strong performance and consistent gains across policies, reward backbones, and benchmarks. This makes the method easier to reproduce, inspect, and extend than pipelines that depend critically on closed-model supervision.
> > >
> > > Accordingly, we do not view the current results as an upper bound of the approach, but rather as evidence that the method is already effective **without requiring proprietary or costly teachers**. We agree that exploring different teacher choices would be valuable future work and could potentially further improve the supervision signal and downstream PRA performance. During the rebuttal period, however, we prioritized the experiments that we believed would be most directly responsive to the reviewer’s remaining concern, namely retraining PRA with **non-Qwen reward backbones**. We hope these additional results are helpful in clarifying the generality of the framework.
> > >
> > > In summary, we agree that the earlier rebuttal did not fully resolve the question of generalizability beyond Qwen on the reward side. We have now addressed that point directly by additionally training PRA with **Llama-3.1-8B-Instruct** and **MedGemma-4B** additionally to **Qwen-3-1.7B** both of which continue to improve the same frozen policy's reasoning over strong baselines. We will include these additional results in the revised version and clarify the discussion of teacher choice to better distinguish what is currently supported by evidence from what remains future work.
> > >
> > > We sincerely thank the reviewer for the careful, thoughtful, and highly valuable feedback. The reviewer’s comments have been very helpful in strengthening the paper, and we hope that the additional experiments and revisions provided here will satisfactorily address the remaining concerns and make the contribution of the work clearer and more compelling.
> > >
> > > Authors of the Submission 24744

---

### Official Review · Reviewer_mFQD · 2026-03-12

**Soundness:** 2
**Presentation:** 3
**Significance:** 2
**Originality:** 3
**Overall Recommendation:** 4
**Confidence:** 4

**Summary:**

This paper studies how to improve knowledge-intensive reasoning at test time without updating the policy model.
To address the medical QA task, the paper proposes *Process Reward Agents* (PRA), a framework that decouples reasoning from retrieval and step evaluation. A frozen policy model generates candidate reasoning steps, while a separate reward agent decides whether to retrieve evidence and assigns step-wise rewards to partial trajectories. These rewards are then used inside beam-search-style decoding to rank and prune candidate reasoning traces online. Experimental results show that combining PRA with Qwen3-4B leads to consistent gains over direct prompting, CoT, RAG, and self-consistency baselines under a matched sample budget.

**Compliance With Llm Reviewing Policy:**

Affirmed.

**Final Justification:**

The author shows that the PRA can improve even the state-of-the-art open-source model. I still believe using PRA in inference time is limited by: (1) will need to enforce the model use specific format to sepearate reasoning steps; (2) unclear PRA is effective or not in tasks beyond the knowledge QA; and (3) explore using PRA to RL the model is more meaningful and interesting. However, the author's rebuttal successfully addressed my previous concerns. Therefore I choose to raise my score but still hold my opinion about the paper's limitation.

**Key Questions For Authors:**

- How do you decompose reasoning traces into individual steps (token level or sentence level)?

- Have you compared using (1) RL with process reward provided PRA to train the model; versus (2) using PRA to perform inference-time beam search?

- Is the beam search with PRA also works for larger LLMs (such as Qwen3-8B or Qwen3-32B)

**Limitations:**

Please see the weaknesses and the questions.

**Strengths And Weaknesses:**

Overall, I found the paper interesting with intuitive motivation and strong empirical results. However, I have several concerns about the strength of portability/generality claims.

**Strengths:**
- The proposed decomposition is conceptually clean. The policy remains frozen, while the PRA handles retrieval and step evaluation. This modularity is attractive, and enables improving the domain-specific reward module without retraining each backbone.

- The reported results are impressive. PRA outperforms CoT, RAG, and their self-consistency variants on MedQA and on several unseen medical benchmarks.

**Weaknesses:**

- The experiments are conducted only in the 4B model. For the large reasoner, whether the capability of small PRA model bottlenecks the performance remains unclear. This raises concerns about the generalization of the pipeline.

- Inference-time beam search with PRA may introduce heavy computational cost, including the GPU memory of deploying the PRA model and the search embedding model. It is necessary to compare such an inference-time beam search with RL training with process rewards.

---

> ### Author Rebuttal · Authors · 2026-03-31
>
> Dear Reviewer mFQD,
>
> We sincerely thank the reviewer for the constructive feedback. Below we address each feedback point by point.
>
> > Question 1: How are reasoning traces decomposed into steps?
>
> We decompose traces at the step level, not at the token or sentence level. Specifically, we prompt the reasoner to produce explicit "Step {number}:" segments. During generation, we use step-specific stop sequences in vLLM, for example the next step header, to stop decoding when a new step begins. We then parse the output with a regular expression matching the step header, and any trailing next-step marker is carried over to the next decoding call. This allows us to segment reasoning into explicit step units cleanly and consistently. This step-based formatting is important because it makes the unit of reward assignment explicit and reproducible, allowing PRA to evaluate semantically meaningful reasoning steps rather than arbitrary token spans.
>
> > Weakness 1 / Question 3: Does Process Reward Agent remain effective for larger policies?
>
> We appreciate the reviewer’s concern and agree that clarifying performance on larger policies is important. Table 2 in the manuscript provides initial evidence that Process Reward Agent (PRA) remains effective beyond the training-policy scale. In particular, a PRA trained with Qwen3-4B-Instruct improves unseen policies across a range of sizes, including the larger reasoner Llama-3.1-8B-Instruct, where accuracy increases from 67.0 to 80.1 (+13.1). To address the reviewer’s concern more directly, we additionally evaluated larger Qwen3 policies. Since there is no Qwen3-8B-Instruct variant, we report both Qwen3-8B and the next larger instruct model, Qwen3-30B-A3B-Instruct. Here, CoT-SC denotes Chain-of-Thought with self-consistency, and RAG-SC denotes Retrieval-Augmented Generation with self-consistency.
>
> |Policy|Method|Acc.|Δ|
> |-|-|-:|-:|
> |Qwen3-30B-A3B-Instruct|CoT-SC@1|85.47|-|
> ||CoT-SC@64|86.48|+1.01|
> ||RAG-SC@64|86.48|+1.01|
> ||**PRA**|**88.05**|**+2.58**|
> |Qwen3-8B|CoT-SC@1|67.31|-|
> ||CoT-SC@64|79.25|+11.94|
> ||RAG-SC@64|73.27|+5.96|
> ||**PRA**|**81.60**|**+14.29**|
>
> For Qwen3-30B-A3B-Instruct, PRA exceeds the best CoT-SC baseline by 1.57 points (88.05 vs. 86.48). For Qwen3-8B, PRA exceeds the best CoT-SC baseline by 2.35 points (81.60 vs. 79.25). In both cases, sampling-based gains begin to saturate, while PRA continues to provide additional improvement.
>
> Importantly, these results do not suggest that PRA capacity saturates before larger policies can benefit. Instead, the same PRA trained on a 4 billion parameter backbone continues to improve both Qwen3-8B and Qwen3-30B-A3B-Instruct, including beyond the strongest self-consistency baselines. This suggests that PRA is not merely compensating for weak policies, but can provide useful step-level control even when the underlying reasoner is substantially larger.
>
> > Weakness 2 / Question 2: Should PRA be compared against reinforcement learning with process rewards, given inference-time cost?
>
> We agree that inference-time cost is an important consideration. A naive implementation of PRA-guided beam search, where each question, beam, and reasoning step is processed independently, would indeed be expensive. In our implementation, however, we use synchronized stage-level batching. Across all active traces, we group pending operations into generation, reward, and retrieval stages, and execute each stage as a single batch regardless of which question or beam a trace belongs to. This substantially improves hardware utilization despite variable reasoning depth, early stopping, and conditional retrieval. We will clarify this implementation detail in Section 3.3 and add pseudocode in the appendix.
>
> We also emphasize a practical efficiency advantage at the model level: the same 4 billion parameter PRA improves much larger unseen reasoners, including Llama-3.1-8B-Instruct, Qwen3-8B, and Qwen3-30B-A3B-Instruct. Thus, the reward model can provide meaningful gains without scaling to match the policy size.
>
> We also note that our goal is different from training a better policy with process rewards. PRA is designed as a reusable inference-time controller for frozen policies, which can be transferred across unseen backbones without retraining the policy itself. In contrast, reinforcement learning with process rewards produces a new policy tied to a particular optimization pipeline, training budget, reward formulation, and stability setting. A rigorous comparison would therefore require a separate study with carefully matched training conditions and compute budgets. We agree that this is an important future direction, but do not view it as a directly matched baseline for the present paper, whose focus is test-time control rather than policy learning.
>
> We thank the reviewer again for the thoughtful and helpful feedback, and we will incorporate these improvements in the revision.

---

> > ### Author Rebuttal · Reviewer_mFQD · 2026-04-03
> >
> > Thank you for the detailed response and for providing the additional experiments and clarifications. These additions make the paper significantly clearer, and I agree that the Process Reward Agent (PRA) represents an interesting new paradigm for test-time control.
> >
> > That said, I still have concerns about its practical significance. First, many of the benchmarks studied in this paper appear to be already close to saturation, even for open-source models. For example, recent work such as OctoMed reports very strong results on knowledge-intensive medical QA benchmarks using Qwen2.5-based models, with 90.8 on MedQA and 72.7 on MedMCQA. This suggests that for knowledge-intensive QA, PRA may not be the most compelling solution in practice, especially when one also considers its additional inference-time cost. More broadly, agentic systems can often query external knowledge bases at inference time, which further reduces reliance on knowledge stored inside the base model and may be a more scalable direction for such settings.
> >
> > Second, while the new results help address the concern about larger policies, they also suggest that the marginal gain of PRA becomes more limited as the policy gets stronger. For instance, on Qwen3-30B-A3B-Instruct, PRA improves over the best CoT-SC baseline by 1.57 points (88.05 vs. 86.48), whereas the gain is larger on Qwen3-8B (+2.35 points, 81.60 vs. 79.25). This pattern seems consistent with diminishing returns on stronger backbones. If the additional benefit shrinks while inference complexity remains nontrivial, the practical value of PRA may be less clear for high-performing models.
> >
> > For these reasons, I still believe that the more valuable research direction would be to explore whether this class of agentic process rewards can play a role during model training, rather than only as an inference-time controller. If such signals can be incorporated into policy optimization or post-training, the contribution would likely have broader and more lasting impact.

---

> > > ### Author Response · Authors · 2026-04-05
> > >
> > > We thank the reviewer for the continued engagement and for noting that our rebuttal helped clarify the paper. Below, we provide additional discussion of the follow-up points regarding diminishing returns on stronger backbones, benchmark saturation, comparison to recent distilled models such as OctoMed, and the role of process rewards during training.
> > >
> > > ---
> > >
> > > > On OctoMed and the practical significance of PRA
> > >
> > > We appreciate the reviewer for highlighting OctoMed as a strong baseline. We believe it is helpful to distinguish the adaptation methods being compared here. **OctoMed is built through a large-scale policy distillation pipeline using GPT-4o and DeepSeek-R1, with millions of reasoning traces and billions of tokens used for fine-tuning.** This requires substantial curation effort and API cost.
> > >
> > > In contrast, **PRA does not fine-tune the policy at all**. It is trained as a separate reward module using only about **40k examples from an open-source teacher**, and is then reused as an inference-time controller for frozen policies. We believe this difference is central to the practical significance of the method.
> > >
> > > The two approaches are also **complementary rather than competing**. To test this directly, we evaluated PRA with **OctoMed-7B itself as the frozen policy**:
> > >
> > > | Method | Accuracy | Δ |
> > > | - | -: | -: |
> > > | OctoMed-7B CoT | 87.4 | - |
> > > | OctoMed-7B CoT-SC@16 | 91.2 | +3.8 |
> > > | OctoMed-7B + PRA | **92.5** | **+5.1** |
> > >
> > > **PRA improves OctoMed-7B by +1.3 points over self-consistency and +5.1 points over chain-of-thought**, despite using only a small reward model and no policy training. We view this as a clear strength of PRA, since it delivers additional performance gains **at substantially lower training cost**, even on top of an already highly optimized distilled policy.
> > >
> > > > On diminishing returns for stronger backbones
> > >
> > > The reviewer observes that the absolute gain of PRA becomes smaller on stronger backbones. We agree that this pattern is visible, but we would interpret it somewhat differently. As base accuracy rises, the remaining error set becomes smaller and harder, so each additional point is more difficult to obtain. In our view, the more relevant question is whether PRA still improves performance after strong ensemble methods have largely saturated, and our answer is yes, including on **OctoMed-7B**.
> > >
> > > > On the claim that the benchmark is saturated
> > >
> > > We believe that, because even the strongest open models still make nontrivial errors, improving these remaining hard cases is precisely where better process-level control matters most.
> > >
> > > More importantly, **our evaluation is not limited to MedQA and MedMCQA**. We evaluate in-distribution performance on the MedQA test split, with all evaluation questions held out from training, and assess generalization on several **unseen datasets**, including **MedBullets, MedMCQA, six MMLU-Med subsets, GPQA, and clinical case datasets from The Lancet and NEJM**. None of these overlap with the MedQA training corpus. Thus, the paper does not rely on a single benchmark, but tests whether PRA improves knowledge-intensive reasoning beyond the training distribution.
> > >
> > > > On comparison to agentic systems with external knowledge access
> > >
> > > We agree that agentic systems with external knowledge access are an important direction. We would also like to clarify that **PRA itself can be viewed as an agentic inference framework**. Our contribution is therefore not positioned against agentic inference, but rather within it: PRA provides a concrete mechanism for process-level control over retrieval-grounded reasoning.
> > >
> > > Our results show that simply enabling retrieval is not the whole story, but how external evidence is incorporated during multi-step reasoning matters substantially.
> > >
> > > > On the suggestion to use process rewards during training
> > >
> > > We agree that using process rewards during policy training is an interesting future direction. This is indeed one of the directions we plan to pursue in future work. At the same time, we would like to clarify that it is a **different problem setting** from the one studied in the present paper. Our contribution here is a **test-time framework for frozen policies**, where a separately trained PRA improves reasoning online **without updating the policy** and transfers across unseen backbones.
> > >
> > > Using process rewards during training would require a policy optimization pipeline with matched choices of optimizer, training budget, reward integration, and stability controls. We therefore view it as a distinct research contribution rather than a missing baseline for the present work.
> > >
> > > We thank the reviewer again for the thoughtful engagement and the positive assessment that our rebuttal clarified the paper substantially. We hope that these additional results and clarifications help further address the reviewer’s follow-up concerns, and we sincerely appreciate the reviewer’s careful consideration.
> > >
> > > Authors of Submission 24744

---

### Decision · Program_Chairs · 2026-04-30

**Decision:**

Accept (regular)

**Comment:**

This paper introduces Process Reward Agents (PRA), a test-time framework that provides online, retrieval-grounded step-wise rewards to frozen reasoning models during beam search decoding. Unlike prior retrieval-augmented PRMs that score completed trajectories post hoc, PRA enables real-time pruning and branching at each generation step.
The reviewers consistently recognize the clean conceptual decomposition of decoupling reasoning from retrieval and verification, the strong ablations isolating the contribution of online process-level control versus post-hoc scoring, and the impressive cross-policy generalization results. During rebuttal, the authors provided substantial additional evidence: scaling to larger policies (Qwen3-8B, Qwen3-30B-A3B), training PRA with non-Qwen backbones (Llama-3.1-8B, MedGemma-4B), evaluation on HotpotQA demonstrating domain transfer, detailed compute efficiency analysis, retrieval sensitivity ablations, and cross-policy calibration results.

Recommendation: Accept